# An Organized Repository of Ethereum Smart Contracts' Source Codes and Metrics

**Giuseppe Antonio Pierro** [1,*] **, Roberto Tonelli** [2,*] **and Michele Marchesi** [2]

1    Inria Lille-Nord Europe Centre, 59650 Villeneuve d'Ascq, France
2    Department of Mathematics and Computer Science, University of Cagliari, 09124 Cagliari, Italy;
     marchesi@unica.it
*    Correspondence: giuseppe.pierro@inria.fr (G.A.P.); roberto.tonelli@dsf.unica.it (R.T.)

**Abstract:** Many empirical software engineering studies show that there is a need for repositories where source codes are acquired, filtered and classified. During the last few years, Ethereum block explorer services have emerged as a popular project to explore and search for Ethereum blockchain data such as transactions, addresses, tokens, smart contracts' source codes, prices and other activities taking place on the Ethereum blockchain. Despite the availability of this kind of service, retrieving specific information useful to empirical software engineering studies, such as the study of smart contracts' software metrics, might require many subtasks, such as searching for specific transactions in a block, parsing files in HTML format, and filtering the smart contracts to remove duplicated code or unused smart contracts. In this paper, we afford this problem by creating *Smart Corpus*, a corpus of smart contracts in an organized, reasoned and up-to-date repository where Solidity source code and other metadata about Ethereum smart contracts can easily and systematically be retrieved. We present Smart Corpus's design and its initial implementation, and we show how the data set of smart contracts' source codes in a variety of programming languages can be queried and processed to get useful information on smart contracts and their software metrics. Smart Corpus aims to create a smart-contract repository where smart-contract data (source code, application binary interface (ABI) and byte code) are freely and immediately available and are classified based on the main software metrics identified in the scientific literature. Smart contracts' source codes have been validated by EtherScan, and each contract comes with its own associated software metrics as computed by the freely available software PASO. Moreover, Smart Corpus can be easily extended as the number of new smart contracts increases day by day.

**Keywords:** ethereum blockchain; solidity programming language; smart contracts; software metrics; corpus

## 1. Introduction

With the advent of blockchain technology as a mainstream technological innovation, many researchers and software developers started investigating the new possibilities for software products relying on such an infrastructure. Second-generation blockchains offer the possibility to code so-called smart contracts in a Turing complete programming language on which all the main operations of traditional software systems can be carried out. The paradigmatic reference is the Ethereum blockchain, which offers the possibility to deploy and execute decentralized applications (dApps) which are mainly coded in Solidity, presently the most adopted programming language [1].

Coding smart contracts which run in a blockchain environment has its peculiarities and constraints and differs from coding in traditional out-of-chain contexts. One of the major differences is the immutability of deployed code: if bugs or bad smells are introduced into a smart contract, these cannot

be fixed afterwards with patches. Another contract must be deployed in substitution of the former and users must be well advised not to use the wrong code. Another main issue is the interaction with the blockchain by means of transactions where information exchange can occur only between blockchain internal components. Furthermore, memory occupation on blockchain typically has a cost that developers want to reduce, and chaining all the blocks poses limitations to the reasonable space available for each smart contract imposing practical constraints to source code size.

This new programming paradigm poses major challenges also for expert developers, and famous failures are commonly found in blockchain software [2,3]. The novelty of the paradigm largely contributes to these faults, since developers do not have historical records or examples to learn from previous code, as it happens in traditional software coding, where software reuse and coding by imitation are reference practices to help in coding better-quality software. Another issue is the lack of reference measures, such as quality, complexity or coupling metrics, which are extensively used in out-of-chain software production to keep software projects under control [4].

The situation is slowly changing for historical records (even if history is quite recent) of software code, since the Ethereum blockchain can now count on up to 1.5 million deployed smart contracts, which have been used and run in the last few years. Access to the source code of this body of smart contracts is still a challenge since transparency and open access granted by public blockchains regards only data registered in the blocks, where only contracts' byte codes are available.

To access the smart contracts' source codes, the developers must resort to other means or to code repositories, such as the classical GitHub or similar resources. Fortunately, in the last few years, EtherScan (https://etherscan.io/) and other web sites have started providing smart-contract checking as a service, so that Ethereum developers can submit their source code to be analyzed and the source code is made available afterwards by the website. However, there is an odd side of the medal for many reasons: access to this body of knowledge is far from easy and far from fast; it is not structured and organized; and the smart-contract metrics are not available and must be computed separately. All these tasks can and need to be automatized to save developers time and work as well as computational resources. Indeed, in the last few years, a number of research papers have been published reporting findings based on smart contracts' source codes, mined from GitHub or some Ethereum block explorer such as EtherScan [5–8]. However, when conducting this kind of empirical research on smart contracts with data from Ethereum blockchain, the abovementioned tasks need to be carried out by the developers themselves. The first task is downloading the smart contracts' source codes to be analyzed. One way to download smart contracts' source code data is to inspect open-source software (OSS) project repositories such as GitHub [9].

Another way to perform this task is to use an Ethereum block explorer. These web services allow users to find the desired information by directly accessing the Ethereum blocks, by using a unique identifier or by sequentially searching several blocks [10]. Some of these Ethereum block explorers provide RESTful Web services, which allow the users to obtain a JSON format payload containing various data. These data may be related to a current or past state of the Ethereum blockchain: an example may be the list of transaction addresses included in a given block of the Ethereum blockchain. This activity might be tedious and time-consuming [11] when conducted by a single user/developer/researcher. Furthermore, the obtained smart contracts' data set can consist of duplicated smart contracts, i.e., smart contracts having different addresses but the same code.

In this work, we tackle these problems and propose an organized, easy to use, large and available software repository for Ethereum smart-contract source codes and metrics where users, researchers, blockchain startups and developers can take advantage of the body of knowledge collected during the last few years. This paper thus proposes Smart Corpus, a repository that provides users with an interface which allows for searching for and downloading smart contracts' source codes. The user interface is available at the following online address: https://aphd.github.io/smac-corpus/. The main challenge of the implementation lies in the fact that the Ethereum blockchain stores a massive amount of heterogeneous data, smart contracts included, which grow enormously in time. For this reason,

Smart Corpus was designed to be scalable by adopting the latest cutting-edge technology, such as document-oriented database, graph query language and serverless computing platform [12].

## 2. Related Work

### 2.1. Previous Literature on Software Corpus Analysis

Gabel and Su [13] built and studied a corpus of open-source software written in three of the most widely used languages: C, C++, and Java. The corpus contains six thousand software projects corresponding to 430 million lines of source code. The authors measured the degree to which each project of the corpus can be "assembled" solely from portions of the corpus, thus providing a precise measure of "uniqueness". Their primary contribution is to provide a quantitative answer to the following question: *how unique is software?* Our work also aims to answer this question because many smart contracts written in the Solidity language have code that is a replication of other smart contracts, although presenting different addresses, as we will see in Section 3.2. Our goal is therefore to answer the question: *how unique are smart contracts written in Solidity?* in order to provide a corpus that is composed of smart contracts which can be distinguished from each other.

Tempero and coauthors [14] presented the "Qualitas Corpus", a curated collection of open-source Java systems. The corpus reduced the time needed to find, collect and organize the necessary source code sets to the time needed to download the corpus. The metadata provided with the corpus explicitly indicate the metrics calculated to identify the main features of the source code: the number of code lines, the number of classes, etc. Our work also aims to present a curated collection of smart contracts equipped with a set of metadata with the aim of allowing experts in the blockchain field to perform static analysis.

### 2.2. Static Analysis on Smart-Contract Code

There is a number of scientific publications with the objective of analysing smart contracts' source codes and testifying to the scientific community's interest in advancing the knowledge on the characteristics of smart contracts' code structure.

Hegedus [15] developed a metric calculator for Solidity code, inspired by the work by Tonelli and collaborators [8]. The metric calculator uses a parser to generate an abstract syntax tree (AST), on which it computes various software metrics, such as the number of code lines for each smart contract, the cyclomatic complexity, the number of functions and the number of parameters for each function. This command-line tool is written in Java and is available on GitHub without license indication since February 2018 (https://github.com/chicxurug/SolMet-Solidity-parser). By using this tool, he calculated and published software metrics results for 10,206 Solidity smart-contract source code files written in Solidity languages. Our work also aims to calculate a set of metadata on the smart-contract corpus by using a similar software.

Pinna and colleagues [16] performed a static analysis on 10,174 smart contracts, deployed in the Ethereum blockchain. The authors showed that some metrics related to smart contracts, such as the number of transactions and the balances, follow power-law distributions. Also, they reported that software code metrics in Solidity have (on average) lower values but higher variance than metrics values in other programming languages for standard softwares. Our work is inspired by their research as Smart Corpus is characterized by (some of) the metrics they defined, as we will explain in Section 3.3.

Pierro and Tonelli [17] pointed out that even the most experienced users, as software developers of smart contracts are, need to be helped to analyse smart contracts and to write a more reliable and secure code. For this reason, an open-source platform (https://aphd.github.io/paso/), called PASO, was proposed as an aid for experts in smart contracts' static analyses. Their work focused on Ethereum blockchain and smart contracts written in Solidity. The platform PASO facilitates debugging of smart contracts by providing software metrics commonly used to comply with coding guidelines.

*2.3. Related Projects*

Other projects similar to Smart Corpus have been previously developed to access online smart contracts' codes deployed in the Ethereum blockchain platform. The projects present specific features and limitations, which are summarized in Table 1.

**Table 1.** Project list, main features and limitations.

| Project's Name | Home Page | REST API URL | Limitations |
| --- | --- | --- | --- |
| GitHub | https://github.com/ | https://developer.git... | Some repositories have restricted access. |
| Ethplorer | https://ethplorer.io/ | https://api.ethplorer... | Requests are limited to 3000/week. |
| EtherScan | https://etherscan.io/ | https://etherscan... | Smart contracts' addresses are not immediately available. |
| EtherChain | https://www.etherch... | https://www.etherch... | Smart contracts' source codes are not available. |
| BlockScout | https://blockscout.com/ | https://blockscout.com... | Smart contracts' source codes are not available. |

### 2.3.1. GitHub

GitHub is the largest collaborative source code-hosting site built on top of the Git version control system [18]. The availability of a comprehensive Application Programming Interface (API) has made GitHub a target for many software engineering and online collaboration research efforts [19]. GitHub offers just open-source software to the community. In GitHub, there are many works regarding projects written in different programming languages, such as Java, Python and Solidity, which is by far the most commonly used language to write smart contracts.

The repository proposed in the paper overcomes the following GitHub limitations:

- The smart-contract source codes collected in GitHub typically do not have a direct reference to smart contracts deployed on the blockchain through an Ethereum address; therefore, it is hard to find out whether it has been really tested or used on the blockchain.
- GitHub does not implement a search engine to filter smart contracts based on particular software metrics, such as the number of modifiers or payables. This is due to the fact that some metrics are specific to the type of language employed to write smart contracts, i.e., Solidity.
- In GitHub, there is no information on smart contracts' use in a real blockchain scenario, on the number of transactions invoking smart contracts or on the number of tokens associated with each smart contract.
- GitHub does not provide smart-contract ABIs or Opcodes.

It is highly probable that the users, especially if they are developers or researchers, want to access smart contracts' source codes, choosing the features implemented in Smart Corpus, on the basis of its specific software metrics and its real usage on the blockchain.

### 2.3.2. Ethereum Block Explorers

Ethereum block explorers are platforms that allow the users to explore and search the Ethereum blockchain for transactions, addresses, tokens and other activities taking place on the Ethereum blockchain [20]. Unlike GitHub, the Ethereum block explorers allow access to only Ethereum data used in the Ethereum blockchain and thus smart contracts' real use cases. To date, in the market, there are different Ethereum block explorers:

- **Ethplorer** (https://ethplorer.io/) provides an API to access many Ethereum data, such as the balances for a specified token and the description of a specific address, but it does not allow

access to the smart contracts' code. The full documentation of the Ethpoler API is available at the following address (https://github.com/EverexIO/Ethplorer/wiki/Ethplorer-API). The requests to API are limited to 5 per second, 50/min, 200/h, 2000/24 h and 3000/week.

- **EtherChain** (https://etherchain.org/) is an explorer for the Ethereum blockchain. Unlike Ethplorer, it claims to provide smart contract code, even though it actually displays the contract byte code and the constructor arguments for a specific smart contract's address. EtherChain provides the API just to access the Oracle gas price predictions (https://www.etherchain.org/api/gasPriceOracle), but not the Ethreum data. If the users want to gather Ethereum data from EtherChain, they need to parse the HTML code.

- **BlockScout** (https://blockscout.com/poa/xdai/) provides an API to access the Ethereum data. It claims to have an API to access only the source code of a few verified smart contracts. Anyway, the addresses list of the verified smart contracts is not available in BlockScout.

- **EtherScan** allows for exploration and searching of the Ethereum blockchain for smart contracts. However, when downloading the smart contracts' source code, the block explorer presents some limitation. First, smart contracts' data and number are huge (on the Giga scale, based on our estimation), but there is a limited API rate of 100 submissions per day per user to retrieve just a smart contract, making the complete download of data an impossible endeavour (https://etherscan.io/apis#contracts). Second, the EtherScan's API does not provide facilities to obtain a list of the smart contracts' addresses, as the existing API calls mainly allow navigation from one block to another. Third, a researcher cannot directly and easily explore the smart contract's source code but, rather, has to first inspect any block in Ethereum and then look for all the transactions that involve an address associated with the smart contract.

## 3. Research Methodology

Smart Corpus has been designed to provide the users with a reasoned repository, i.e., a repository which is not just a webspace where to collect them but also mainly a service to help the researchers filter and analyze the smart contracts' source codes. To this aim, Smart Corpus has been planned to perform four main automatic operations on smart contracts' source codes (data):

1. data retrieving,
2. data cleaning,
3. data modelling and
4. data querying.

Figure 1 shows the Smart Corpus's pipeline of operations.

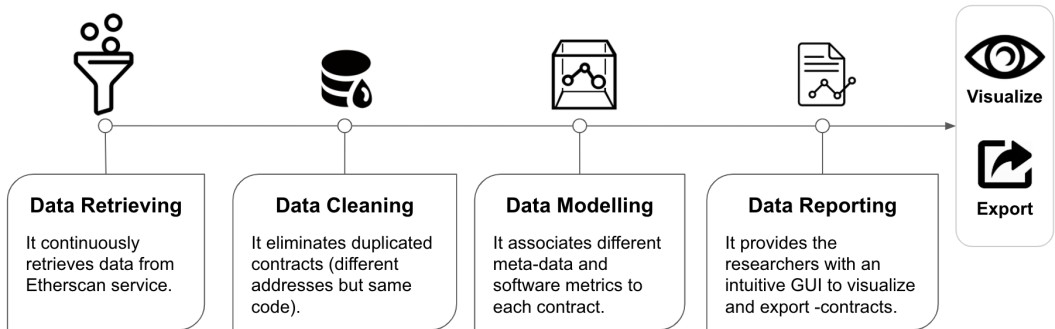

**Figure 1.** Smart-Corpus's pipeline model.

### 3.1. Retrieving Data

We collected smart contracts' source codes, smart contracts' application binary interfaces (ABIs) and smart contracts' byte codes through the EtherScan website, which makes available the source code

of a subset of verified smart contracts deployed on the Etehreum blockchain, though in a labourious way. We instead made this task easier and automatic via a retrieving data script available at the following address (https://github.com/aphd/solidity-metrics/tree/master/examples). During this phase, the blockchain blocks are automatically inspected. Each block is formed by a list of transactions between two different blockchain addresses, which can refer to a wallet or to a smart contract. The script looks for addresses that refer to a smart contract, and when the source code is available, it downloads the smart contract's source code, the ABI and the byte code. The data coming from the source code are not immediately available as they are embedded in the HTML code of the webpage provided by EtherScan. Therefore, the script removes the HTML tags and stores the code cleaned up.

Figure 2a shows how Smart Corpus finds the smart contracts' list in a given block. Figure 2b shows the HTML page where the smart contract code is available. The HTML page containing the smart-contract code and the HTML tags is downloaded. Figure 2c shows the HTML code that will be processed to remove the HTML tags and to save just the Solidity source code of the smart-contract.

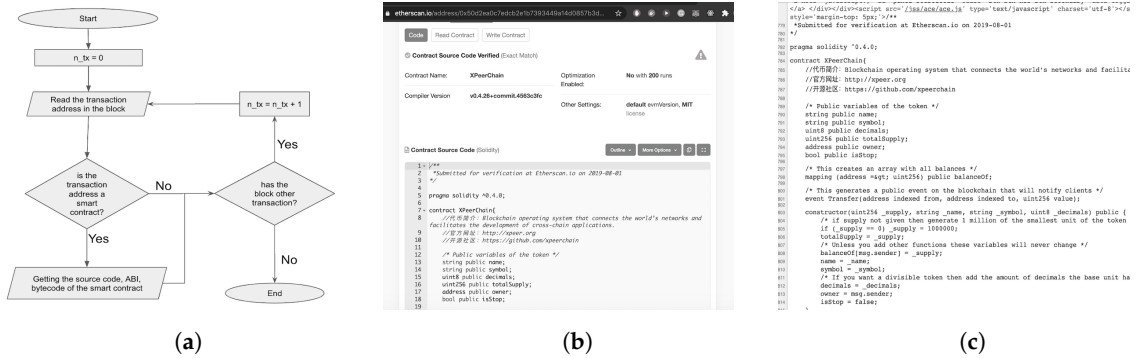

| (**a**) | (**b**) | (**c**) |

**Figure 2.** Data retrieving pipeline: Figure 2a–c shows three different phases to retrieve the smart contracts. (**a**) Transactions list in a block, (**b**) smart contract's webpage code and (**c**) smart contract's source code.

The smart contracts' codes are stored in the filesystem of the Smart Corpus server. Due to the quota limits on queries per second (the EtherScan website allows a few connections per second), Smart Corpus contains only a portion of all available smart contracts. However, the retrieving data phase is continuously collecting data, starting from 10 December 2019. To date, thirty thousand smart contracts (source code, ABI and byte code) have been downloaded and made available through Smart Corpus.

### 3.2. Cleaning Data

Each smart contract in the Ethereum blockchain is distinguished from any other smart contract as it is identified by a unique address, i.e., a hash of 160 bits, and its byte code is stored on the blockchain [21]. Indeed, each time a smart contract is deployed in the network, either in the main or in the test network, a unique address is associated with the smart contract even in the case where the source code of two or more smart contracts is the same. However, this is a problem for the analysis of the software metrics because the smart contracts are distinguished only on the basis of their address and not on their content. Therefore, Smart Corpus eliminates double contracts in order to provide a clean smart contracts' corpus on which to perform the analysis. To this aim, duplicate smart contracts have been defined on the basis of their content, i.e., having the same code despite presenting different addresses.

### 3.3. Modelling Data

Unlike the tools discussed in the related work of Section 2, Smart Corpus associates different metrics (intrinsic metrics and extrinsic metrics) to the smart contracts, aiming to facilitate the selection of

a smart-contract set that meets precise requirements. The metrics associated with the smart contracts are then stored in a document-oriented database. Figure 3 shows the database schema of a smart contract.

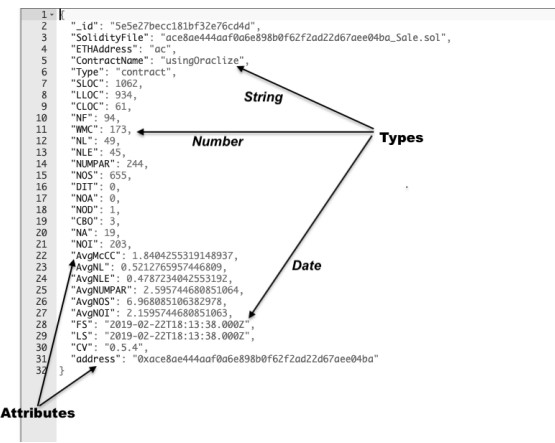

**Figure 3.** Smart Corpus's database schema.

### 3.3.1. Smart Contracts' Intrinsic Metrics

The smart contracts' intrinsic metrics are smart contracts' software metrics which depend on internal properties of the smart contracts' code, such as the number of lines of code, modifiers, payable, etc. Table 2 shows the smart contracts' intrinsic software metrics.

**Table 2.** Smart contracts' intrinsic metrics.

| Name | Description |
| --- | --- |
| Pragma | "Pragma" indicates which version of Solidity compiler is used to prevent issues with future compiler versions. |
| SLOC | "SLOC" indicates the number of lines in a smart contracts' source code. |
| Modifiers | "Modifiers" indicates the number of function modifiers in a smart-contract. |
| Payable | "Payable" indicates the number of payable functions in a smart-contract. |
| Mapping | "Mapping" indicates the number of variables of mapping types in a smart-contract. |
| Address | "Address" indicates the number of variables of address types in a smart-contract. |

### 3.3.2. Smart Contracts' Extrinsic Metric

The smart contracts' extrinsic metrics are properties depending on external factors rather than the code itself, such as the number of transactions executed to the smart contracts or the number of tokens associated with the smart contracts. Table 3 shows the smart contracts' extrinsic metrics.

**Table 3.** Smart contracts' extrinsic metrics.

| Name | Description |
| --- | --- |
| Transactions | "Transactions" represent the total number of transactions generated by the smart contract (sent or received). |
| Balance | "Balance" is the amount of crypto coins associated with a smart-contract address. |
| EtherValue | "EtherValue" is the dollar value associated with a smart-contract address. |
| Token | "Token" is the value for each token associated with a smart-contract address. |
| Last_seen | "Last_seen" is the timestamp of the last time that the smart contract was used (sent or received). |
| First_seen | "First_seen" is the timestamp of the first time that the smart contract was used (sent or received). |

*3.4. Filtered Data*

The smart contracts' source code is stored in a file system and is organized in folders and subfolders to ease the navigation. Figure 4 shows the subdirectory structure. The first leaf corresponds to the first two letters of the smart-contract address, and then, each directory contains the file named using the full address of the smart contract and three different extensions, respectively .sol for the Solidity source code, .abi for the ABI and .bytecode for the byte code.

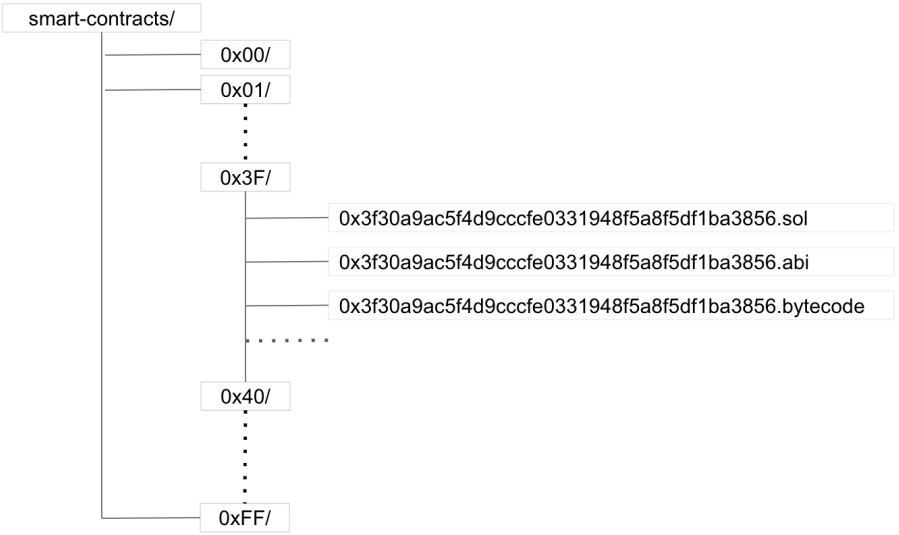

**Figure 4.** Smart contracts' directory structure.

The metadata (both the intrinsic and extrinsic metrics) are stored in a document-oriented database: MongoDB [22]. The choice to use a document-oriented database instead of a relational database such as Mysql is based on the following:

- Relational databases are prone to deterioration when data sets overcome a size threshold, while a document-oriented database such as MongoDB comes with an inbuilt load balancer, which makes it a better solution in applications with high data load [23]. We update MongoDB each day to generate the data archive.
- Unlike relational databases where data is stored in rows and columns, document-oriented databases store data in documents. The documents typically use a structure similar to JSON (JavaScript Object Notation); they indeed provide a natural way to model data that is closely aligned with object-oriented programming. Each document is considered an object in

object-oriented programming; similarly, each document is a JSON in document-oriented database. The concept of a schema in document databases is dynamic: every document might contain a different number of fields. This is useful when modeling unstructured and polymorphic data. Also, document databases allow robust queries: any combination of fields in the document can be combined for querying data [24].

*3.5. User Interface*

Smart Corpus' graphical user interface (GUI) allows users to access the smart contracts' repository. There are two ways to access the smart contracts' repository: through the "HTML user interface" and through a "GraphiQL application", both of them via a web browser.

3.5.1. Smart Corpus HTML User Interface

The Smart Corpus HTML user interface is publicly available since January 2020 (https://aphd.github.io/smac-corpus/). Figure 5 shows the different components of the GUI.

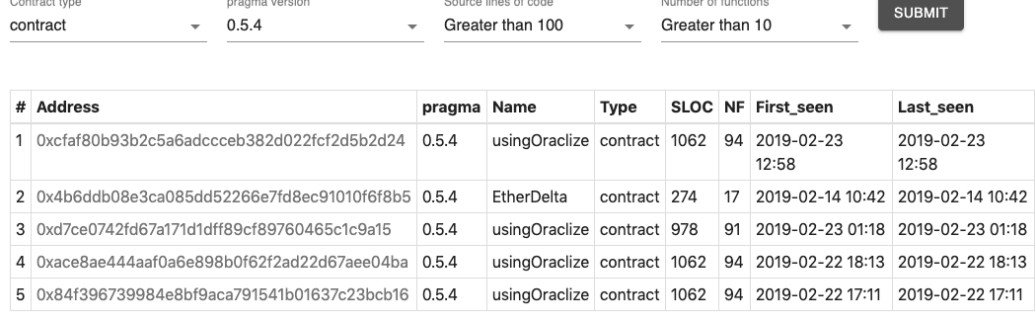

**Figure 5.** Smart Corpus's user interface.

- At the top, the user can find the form to filter the smart contracts. The form is made of a number of drop-down lists, each one corresponding to a different metric and a submit button to perform the research. The GUI form allows the user to inspect smart contracts based on some metadata, such as the "pragma version", and software metrics, such as the numbers of "modifiers" and/or the numbers of "payable".
- Below the form, the smart contracts filtered by the user are displayed. For readability, only a part of the smart-contract metrics are presented in the table layout format. Each column header in the table indicates the name of a metric associated to smart contracts. While the HTML GUI displays just some metrics, the user can access all the metrics and the smart contracts' source codes by selecting the checkbox displayed on the right of the smart-contract address and by clicking on the red button "download". The user can also access the original repository where the smart contract was retrieved, i.e., the EtherScan service.

3.5.2. Smart Corpus GraphQL Application

Graph Query Language (GQL) is a full data query language to implement web-based services, centered on high-level abstractions, such as schemas, types, queries and mutations. GQL is a domain-specific language internally developed in Facebook from 2012 onward and publicly announced in 2015, with the release of a draft language specification. The language was conceived with the following goals:

- To reduce possible overload of data transfer relative to Representational State Transfer (REST)-like web service models in terms of both the amount of data unnecessarily transferred and the number of separate queries required to do it.

- To reduce the potential of errors caused by invalid queries on the part of the client. In particular, with the GQL application, the user can execute "type introspection", i.e., the user can examine the type or properties of an object at runtime. For example, thanks to introspection queries, the user can find out both the intrinsic and the extrinsic metrics associated with a specific smart-contract while typing the query.

Figure 6 shows an example query and its result.

**Figure 6.** Example use of variables to filter a query result with GraphQL.

Smart Corpus GQL application, unlike the Smart Corpus HTML user interface, is still in the development and testing phase. However, the Smart Corpus GQL application source code is publicly available and can be downloaded and deployed on any platform having the software requirements specified in its documentation (https://github.com/aphd/smac-corpus-api). In the Appendix A we present all the queries GraphQL application can perform.

*3.6. Use Case*

A use case for Smart Corpus might concern a researcher interested in the static analysis of smart contracts. For example, the researcher might be interested in performing an analysis of smart contracts written with a particular version of the Solidity language, 6.0, and having at least a payable function in the smart contract. The research of smart contracts that meets these requirements would be very expensive in terms of time, work and computational resources using a service like EtherScan. Instead, thanks to Smart Corpus, the user needs to perform only a few steps, as described below:

- connect to the service through the link: https://aphd.github.io/smac-corpus/,
- select the option "version 6.0" from the drop-down menu entitled "pragma version",
- select the option "greater than zero" from the drop-down menu entitled "number of payables" and
- submit the form by clicking on the button "submit".

After few seconds, depending on the number of smart contracts that meet the requirements specified by the user, the smart contracts' addresses and the metrics values will be displayed in a table layout format ready to be downloaded.

## 4. Results

Smart Corpus has been in use for 10 months, since December 2019, and 100 K smart contracts have been downloaded via the user interface. Until the paper was written (October 2020), Smart Corpus was a curated corpus of 30 K smart-contract source codes, ABI and byte codes with related metadata and software metrics. As time passes, Smart Corpus is continuously increasing at a rate of 100 smart contracts per day. Figure 7 shows the number of smart contracts' source codes, ABI and byte codes

retrieved per day since Smart Corpus was deployed for the first time. For each smart contract, Smart Corpus computed extrinsic and intrinsic metrics, as described in Sections 3.3.1 and 3.3.2.

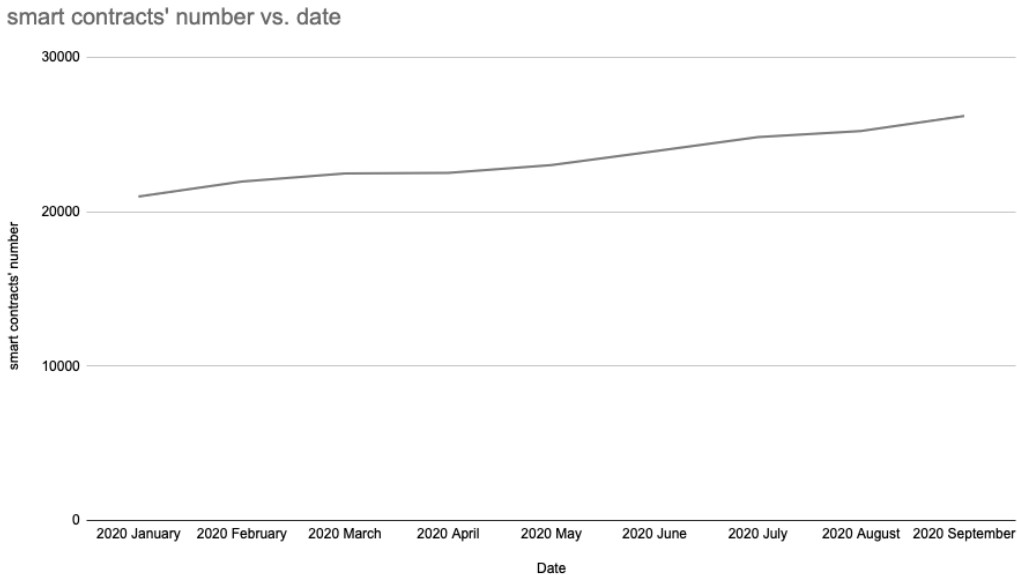

**Figure 7.** Number of smart contracts collected in Smart Corpus.

Summing up, Smart Corpus has two GUIs to access data: the HTML GUI and the GraphQL interface. The HTML GUI is described in Section 3.5.1, while the GraphQL interface is described in Section 3.5.2. The GraphQL interface gives blockchain researchers the ability to request for exactly what they need. The user can directly access the results via GraphQL interface, as shown in Figure 6.

Unlike the existing repositories (see Section 2.3.2) which make available the source code in a laborious way, Smart Corpus instead made this task easier and faster. Indeed, one of the advantages of using Smart Corpus lies in the fact that it can reduce the costs in performing the smart-contract static analysis. For example, it can be used to easily analyze design and programming patterns for the smart-contract programming language.

Even though the Smart Corpus service has been working for a few months and has not been advertised yet, it has already collected 30K smart contracts, thus providing an interesting and helpful future venue for researchers and software developers interested in the blockchain. Moreover, Smart Corpus allows for analysis of how industry companies use the Solidity programming language to solve concrete problems in different application areas, such as healthcare, insurance, transportation, government, entertainment and energy.

## 5. Conclusions and Future Works

In this paper, we described the Smart Corpus project, an effort to bring smart-contract data (source codes, ABIs and byte codes) to the hands of the research community, providing help to reproducible research and a less time-consuming way to gather data and to perform static analysis. The project has already stored several megabytes of data, which correspond to about thirty thousand smart contracts. This work corresponds to 10 months of data retrieving that are made available to the blockchain scientific community and blockchain developers in a few seconds. The Smart Corpus data set has strong potential to provide an interesting venue for research in many software engineering areas, including but not limited to the best practices for Solidity software development, distributed collaboration, and code paternity and attribution. The Smart Corpus project is in its initial stage of development, but it can already provide useful insight for researchers on smart contracts' coding and everyday use in the blockchain. The corpus will continue to be expanded in content and in the

provision of intrinsic and extrinsic metrics, thus becoming more and more representative of the Solidity code actually used in the blockchain community.

**Author Contributions:** Conceptualization, G.A.P. and R.T.; Data curation, G.A.P.; Formal analysis, G.A.P.; Investigation, G.A.P. and R.T.; Methodology, G.A.P.; Project administration, R.T.; Software, G.A.P.; Supervision, R.T. and M.M.; Validation, G.A.P.; Visualization, G.A.P.; Writing—original draft, G.A.P.; Writing—review & editing, G.A.P. and R.T. All authors have read and agreed to the published version of the manuscript.

**Funding:** This work was partially funded by the CRYPTOVOTING project, financed by the Sardinia Region, call POR FESR Sardegna 2014–2020, Prot. 0010083, n. 1361 REA, 01/08/2018, and by the ABATA project (Application of Blockchain to Authenticity and Traceability of Aliments), funded by the Italian Ministry for Economic Development, National Operational Program "Enterprises and Competitiveness", project n. F/200130/01-02/X45; Fondazione di Sardegna, year 2019, grant. n. F72F20000190007.

**Conflicts of Interest:** The authors declare no conflict of interest.

## Abbreviations

The following abbreviations are used in this manuscript:

| | |
|---|---|
| ABI | The smart-contract Application Binary Interface |
| GQL | Graph Query Language |
| JSON | JavaScript Object Notation |
| SLOC | Source lines of code |
| REST | Representational State Transfer |

## Appendix A. Queries

Listing A1 displays a GQL query that returns smart contracts' addresses having more than 20 methods defined in a contract. Listing A2 displays the query results in the JSON format. The query output, in addition to the smart contract's addresses, contains various information (intrinsic metrics) such as the number of events, the number of functions, the number of modifiers and the number of payables, as specified by the query A1,

Listing A1: A Graph Query Language (GQL) query for displaying intrinsic metrics.

```
{
    metrics(query:{functions_gt: 20}) {
        adress
        events
        functions
        modifiers
        payable
    }
}
```

Listing A2: A GQL result displaying intrinsic metrics.

```
{
    "data": {
        "metrics": [
            {
                "contractAddress": "0xb7f4c286851cbf0cbf2fe8ebf40412b196c0e8ad",
                "events": 7,
                "functions": 27,
                "modifiers": 1,
                "payable": 1
            },
            {
```

```
        "contractAddress": "0x755cebe8cc53c7cb1e1bb641026a17d37d4aea91",
        "events": 4,
        "functions": 31,
        "modifiers": 1,
        "payable": 4
    },
    {
        "contractAddress": "0xb92aa4a864daf0d6a509e73a9364feba44384965",
        "events": 3,
        "functions": 24,
        "modifiers": 1,
        "payable": 1
    },

    ...

    }
}
```

Listing A3 displays a GQL query that returns some extrinsic metrics of a specific smart contract's address. Listing A4 displays the query results in the JSON format. The query output, in addition to the smart contract's address, contains information such as the total number of transactions generated by the smart contract and the amount of crypto coins associated with the smart contract's address specified by the query A3,

Listing A3: A GQL query for displaying exstrinsic metrics.

```
{
    metrics(query:{address_eq: "0x536c7efeebff067a69393133b1c87a163a6b0598"})
    {
      adress
      transactions
      balance
    }
}
```

Listing A4: A GQL result displaying exstrinsic metrics.

```
{
    "data": {
      "metrics": [
        {
          "contractAddress": "0x536c7efeebff067a69393133b1c87a163a6b0598",
          "transactions": 639 ,
          "balance": 0 Ether
        }
      ]
    }
}
```

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
