# Peer review of "An Organized Repository of Ethereum Smart Contracts’ Source Codes and Metrics"

_futureinternet, doi:10.3390/fi12110197_

Round 1
Reviewer 1 Report
Overall the presentation of the paper is really nice. I have some comments for further improvement of the manuscript. This paper proposes a repository for Erhereum based smart contract code. Could authors please justify in more detail why there is a need to propose a repository when Github is already there to access the code. Could you please also make comparison between your proposed repository and GitHub. What are the significant contributions of this work for the scientific community.
Author Response
We thank the reviewer for his/her positive feedback and constructive comments. Below we respond to the specific/technical comments point-by-point.
Overall the presentation of the paper is really nice. I have some comments for further improvement of the manuscript. This paper proposes a repository for Erhereum based smart contract code.
Could authors please justify in more detail why there is a need to propose a repository when Github is already there to access the code. Could you please also make comparison between your proposed repository and GitHub.
Reply: Thanks for the question, which is very relevant. GitHub is indeed the largest collaborative source code hosting site, but we think that it cannot satisfy particular specificities of the smart-contract programs. First of all, GitHub does not implement a search engine to filter smart-contracts based on particular software metrics such as the number of modifiers or payables. This is due to the fact that some metrics are specific only to the type of language employed to write smart-contracts, i.e. Solidity. Furthermore, in the case of smart contracts deployed on GitHub there is no information on their use in a real blockchain scenario, on the number of transactions invoking a smart-contracts or the number of tokens associated with the smart contracts. All these types of information might be relevant and helpful for users.
In particular, GitHub is not a repository of smart contracts and even when GitHub software projects contain smart contracts code there is no guarantee that these are properly working or have been really deployed and used in the Ethereum main net. Even when deployed and used, GitHub does not provide the address of deploy, the date, the block and other information typically associated with a smart contract. We can basically say that smart contract code on GitHub does not automatically represent a real smart contract. Real smart contracts are only the ones residing in a blockchain. Furthermore, it does not provide smart contracts ABI or Opcode and it is not possible to automatically query GitHub to collect a group of smart contracts but users must explore the single repositories to recover the code.
We added this comparison in the paper at the lines 142-155.
What are the significant contributions of this work for the scientific community?
Reply: Thanks for the question, which lets us make Smart-Corpus’ significance explicit. Even though Smart-Corpus service has been operational for a few months and has not been advertised yet, it has already collected 30K smart-contracts, thus providing an interesting and helpful future venue for researchers and software developers interested in the Blockchain. We think indeed that Smart Corpus service has strong potential to provide an interesting venue for research also in more specific software engineering areas including, but not limited to, the best practices for Solidity software development, distributed collaboration, and code paternity and attribution.
We explained at lines 330-335 why this work provides a significant contribution for the scientific community interested in blockchain.

Reviewer 2 Report
General Comments/Suggestions
Question:
How's the design for security and integrity of data in the Smart-Corpus?
Line 11
organized reasoned and up to date repository ...
=> organized, reasoned, and up-to-date repository ...
Line 321
unlike the existing repositories (see Section 2.3.2) which makes available ...
=> unlike the existing repositories (see Section 2.3.2) which make available ...
Author Response
We thank the reviewer for his/her positive feedback and constructive comments.
Below we respond to the specific/technical comments point-by-point.
Question: How's the design for security and integrity of data in the Smart-Corpus?
Reply: Thanks for the question. We managed to have a safe service allowing the external user to perform only HTTP GET requests to the repository. Mostly, the users are just allowed to retrieve smart-contracts source code or/and the related metrics via an HTTP GET method request. This means that the server cannot accept any POST request which is meant to change the state of the repository. Furthermore, we set a limit on the number of requests per seconds the server can accept (10 / seconds), to prevent our GET API from being overwhelmed by too many requests. In the future, if the requests will increase, the limit of requests could be raised, going towards a new architecture client-server, such as the serverless lambda architecture.
Line 11 organized reasoned and up to date repository … => organized, reasoned, and up-to-date repository …
Reply: Thanks, we amended the error.
Line 321
unlike the existing repositories (see Section 2.3.2) which makes available ...
=> unlike the existing repositories (see Section 2.3.2) which make available …
Reply: Thanks, we amended the error.
